# Effect of Light Quality on Seed Potato (*Solanum tuberose* L.) Tuberization When Aeroponically Grown in a Controlled Greenhouse

**DOI:** 10.3390/plants13050737

**Published:** 2024-03-06

**Authors:** Md Hafizur Rahman, Md. Jahirul Islam, Umma Habiba Mumu, Byeong Ryeol Ryu, Jung-Dae Lim, Md Obyedul Kalam Azad, Eun Ju Cheong, Young-Seok Lim

**Affiliations:** 1Department of Bio-Health Convergence, College of Biomedical Science, Kangwon National University, Chuncheon 24341, Republic of Korea; hafizknu94@gmail.com (M.H.R.); ummemumu0@gmail.com (U.H.M.); fbqudfuf0419@naver.com (B.R.R.); ijdae@kanwon.ac.kr (J.-D.L.); 2Bangladesh Agricultural Research Council (BARC), Crops Division, Farmgate, Dhaka 1215, Bangladesh; jahirulislam213@gmail.com; 3Department of Chemistry and Biochemistry, Food and Dairy Innovation Center, Boise State University, Boise, ID 83725, USA; azadokalam@gmail.com; 4College of Forest and Environmental Science, Kangwon National University, Chuncheon 24341, Republic of Korea; 5LYS Potato Research Institute, College of Bio-medical Science, Building A, 303, Kangwon National University, Chuncheon 24341, Republic of Korea

**Keywords:** artificial light spectrum, potato tuberization, tuber number, growth characteristics, photosynthetic activity

## Abstract

A plant factory equipped with artificial lights is a comparatively new concept when growing seed potatoes (*Solanum tuberosum* L.) for minituber production. The shortage of disease-free potato seed tubers is a key challenge to producing quality potatoes. Quality seed tuber production all year round in a controlled environment under an artificial light condition was the main purpose of this study. The present study was conducted in a plant factory to investigate the effects of distinct spectrum compositions of LEDs on potato tuberization when grown in an aeroponic system. The study was equipped with eight LED light combinations: L1 = red: blue: green (70 + 25 + 5), L2 = red: blue: green (70 + 20 + 10), L3 = red: blue: green (70 + 15 + 15), L4 = red: blue: green (70 + 10 + 20), L5 = red: blue: far-red (70 + 25 + 5), L6 = red: blue: far-red (70 + 20 + 10), L7 = red: blue: far-red (70 + 15 + 15), L8 = red: blue: far-red (70 + 10 + 20), and L9 = natural light with 300 µmol m^−2^ s^−1^ of irradiance, 16/8 h day/night, 65% relative humidity, while natural light was used as the control treatment. According to the findings, treatment L4 recorded a higher tuber number (31/plant), tuber size (>3 g); (9.26 ± 3.01), and GA_3_ content, along with better plant growth characteristics. Moreover, treatment L4 recorded a significantly increased trend in the stem diameter (11.08 ± 0.25), leaf number (25.32 ± 1.2), leaf width (19 ± 0.81), root length (49 ± 2.1), and stolon length (49.62 ± 2.05) compared to the control (L9). However, the L9 treatment showed the best performance in plant fresh weight (67.16 ± 4.06 g) and plant dry weight (4.46 ± 0.08 g). In addition, photosynthetic pigments (Chl *a*) (0.096 ± 0.00 mg g^−1^, 0.093 ± 0.00 mg g^−1^) were found to be the highest in the L1 and L2 treatments, respectively. However, Chl *b* and TCL recorded the best results in treatment L4. Finally, with consideration of the plant growth and tuber yield performance, treatment L4 was found to have the best spectral composition to grow quality seed potato tubers.

## 1. Introduction

Potatoes (*Solanum tuberosum* L.) are the world’s fourth most widely grown food crop, behind wheat, rice, and maize, yielding 400 million tons annually [1]. Potato is a well-known source of carbohydrates, proteins, minerals, and vitamins that is grown in over 100 countries and provides food for more than a billion people [2]. As a result, increasing the output of this root crop could be a crucial point for meeting nutritional needs and the demand for an ample supply of high-quality seeds for reproduction.

Potatoes are an annual dicotyledonous herbaceous tuber crop that thrives in chilly temperate climates with plenty of sunshine, moderate daily temperatures, and cool nights. The tubers are underground shoots, and their thickening necessitates a large photosynthetic sink [3]. Potato plants are prone to viruses that significantly deteriorate the germplasm and reduce the yield and quality [4]. The production of pathogen-free in vitro plant materials is a current demand that effectively removes virus infection from microtubers and minitubers and secures the yield. To cultivate quality minitubers, a controlled environment with natural or artificial light conditions is needed [5]. Light (intensity, photoperiod, and spectrum) is an essential abiotic environmental element that regulates the photosynthesis, metabolism, and morphogenesis of potato plants. It produces energy for photosynthesis and signal cues for physiological functions [6]. Pre-basic seeds, also known as the nucleus seeds or breeder’s seeds, are propagated in aeroponics or controlled environmental conditions under a rigorous management system [7,8]. Potato pre-basic seed tuber production in artificial light conditions (plant factory conditions) is a relatively new concept for producing seed tubers disregarding seasonal commitments all year round [9]. Following the establishment of a plant factory, a controlled environment for plant growth in a simulated environment maintains phenotypic durability and yield enhancement [10].

Vigorous axillary branches and an increased leaf surface with a higher chlorophyll content in potato plants have been observed under red and far-red light combinations [11]. In another study, the addition of green light with blue and red spectra was found to be effective in producing plant biomass. Green light penetrates the plant canopy better than blue or red light [12,13,14], and that is why the addition of green light to blue and red light promotes plant growth by allowing leaves in the lower canopy to use the transmitted green light in photosynthesis [14,15]. The wavelengths of red and blue light, which stimulate photosynthetic biosynthesis, are effective in increasing physiological and photosynthesis activity [16,17]. Even though red, blue, and green light all promote potato growth, few studies have looked at the effects of different LED spectra on the tuber biomass allocation [18].

Therefore, the present experiment was undertaken to observe the effect of different LED spectral combinations on the morpho-physiological response and tuber seed yield in potato under a plant factory system.

## 2. Materials and Methods

### 2.1. Plant Materials

Professor Young-Seok Lim at Kangwon National University, a breeder of this variety, provided the ’Happy King’ (also called as ‘Unification’ and N-198). The mother plant was multiplied in vitro in an aseptic condition under artificial white LED light (Apack, INC, Daejun, Republic of Korea) with a photosynthetic photon flux density (PPFD) of 100 mol m^−2^ s^−1^ in a plastic culture vessel (8 cm × 12 cm; SPL Life Sciences Co., Ltd., Pocheon-si, Republic of Korea). The photoperiod, relative humidity (RH), and in vitro growth room temperature were set to 16/8 (day/night), 70%, and 25 °C, respectively. The 30-day-old plantlets were transplanted straight into plastic plant-growing 16 plug-in trays (30 cm × 30 cm) and placed in a greenhouse covered with a Sunshade Clothe 70% Sunblock Black Greenhouse Shade Net (QINGDAO SUNTEN PLASTIC Co., Ltd., Shandong, China). Acclimatization took 10 days, maintaining the relative humidity (RH) at 70% and temperature at 25 °C. Following the acclimatization period, the plants were transferred to an aeroponic cultivation bed in a controlled greenhouse.

### 2.2. Plant Growing Conditions and Artificial Light-Emitting Diode (LED) Compositions

The virus-free potato (*Solanum tuberosum* L.) plantlets (tested by ISK 20001/0025, Agdia, Inc., Elkhart, IN, USA; Figure 1) were transplanted (Fall season; September–November 2021) to a steel-made chamber structure (80 cm × 60 cm × 80 cm) covered by a black curtain (QINGDAO SUNTEN PLASTIC Co., Ltd., Qingdao, China), with eight LED light (Bisol LED Light Co., Seoul, Republic of Korea) combinations (Table 1). The photosynthetic photon flux density (PPFD) was 300 µmol m^−2^ s^−1^, the temperature 18–25 °C and the photoperiod was 16/8 h day/night (6.00 to 8.00 p.m.).

### 2.3. Aeroponic System

The aeroponic system was built with an aluminum frame and expanded foam tray panels. The nutritive solution was continuously pumped from the supply reservoir. Solutions A and B and the nutrient formula followed by [19] were stored in tanks A and B, respectively, and were mixed in a mixing tank before being transferred to the supply tank with the EC (1.2–1.7 dS m^2^) and pH (5.5–6) adjusted. The nutrient solutions were mixed and delivered to the supply tank automatically, and they were changed and cleaned once a week. To moisten the plant roots, micro-sprinklers (Naan Dan Jain Irrigation System, Ltd., Tel Aviv, Israel) were used to continuously supply a nutritional solution that was circulated by several pumps in the tubing network for 10 s at a time, with 2 min intervals between mists. The leftover nutrient solution was returned to the reservoirs and recirculated. Throughout the experiment, the quality of the nutrition solution (EC and pH) was checked daily. To maintain the pH, HCl (1 N) and NaOH (5 M) were used.

### 2.4. Morphological Traits Analysis

Matured potato plant (70 DAT) morphological trait data were recorded (three plants sample/treatment), including the stem diameter, shoot length, leaf number, leaf length, leaf width, branch number, stolon length, root length, plant fresh weight, and dry weight. The shoot length, leaf number, leaf length, leaf width, root length, and stolon length data were taken by a meter scale, the shoot length was taken from the bottom of the plant to the top leaf, and the leaf length and width were taken from the third leaf from the top leaf. The stem diameter data were taken from the last one-third of the main stem with a digital caliper (Digital caliper Guanglu, 0–100 mm), and number of all the branches was counted. The whole plant fresh weight and dry weight were taken with a digital scale (Citizen CY 220) with an accuracy of 0.001. Then, the whole plant was dried in an oven (Model OF-12GW, JEIO TECH, Daejeon, Republic of Korea) at 60 °C for 72 h and the dry weight recorded.

### 2.5. Leaf Gas Exchange Measurement

The photosynthetic data were taken 40 days after transplantation. The net photosynthetic rate (A, µmol m^−2^ s^−1^), transpiration rate (E, mmol m^−2^ s^−1^), stomatal conductance (gs, mmol m^−2^ s^−1^) were measured on well-developed leaves (third leaf from the top) of six plants under each treatment using an LCpro gas analyzer (ADC BioScientific Ltd., Hoddesdon, Herts EN11 ONT, UK). The levels of A, gs, E, and WUE were measured at the ambient environmental condition. The measurements of the gas exchange were carried out at mid-day between 10.00 a.m. and 3.00 p.m. The photosynthetic water use efficiency (WUE) was calculated as the ratio A/E.

### 2.6. Analysis of Photosynthetic Pigments

The photosynthetic pigments of the potato plants were investigated, including chlorophyll *a* (Chl *a*), chlorophyll *b* (Chl *b*), total chlorophyll (TCL), and carotenoid (40 DAT). Three plant leaf samples from each treatment were taken for the photosynthetic pigment analysis. Already harvested leaves were immediately immersed in liquid nitrogen and preserved at −80 °C for subsequent study. Fresh (500 mg) leaves were macerated in acetone (10 mL) with the use of a mortar and pestle and left at room temperature for 15 min to detect the photosynthetic pigments. The extracted material was placed in a tube and centrifuged at 5000 rpm for 10 min. The absorbance was measured using a spectrophotometer at 647, 663, and 470 nm (UV-1800 240 V, Shimadzu Corporation, Kyoto, Japan). The photosynthetic pigments were estimated using Lichtenthaler’s [20] formula and reported in milligrams per gram of fresh weight (FW).
Chl *a* = 12.25 × A_663_ − 2.79 × A_647_
Chl *b* = 21.50 × A_647_ − 5.10 × A_663_
Car = [(1000 × A_470_) − (1.82 × Chl *a*) − (85.02 × Chl *b*)]/198
TCL = 7.15 × A_663_ + 18.71 × A_647_

### 2.7. SPAD Index Analysis

The SPAD readings were taken 40 days after transplantation in the pot using a hand-held chlorophyll meter model SPAD-502 Plus (KONICA MINOLTA, Thomas, VI 00802-6430, USA). Before taking the readings, the instrument was calibrated with the reading checker according to the recommendations in the manual. Ten plants per plot were sampled, and readings were taken in the terminal leaflet of the fourth leaf fully expanded from the apex of the plant, avoiding reading in the central vein of the leaflets. When the potato plant did not have the fourth leaf fully expanded yet, the oldest leaf was chosen for the SPAD readings. Care was taken not to sample unhealthy (pest attack and disease) and out-of-spacing plants. The SPAD readings were performed during the morning (8:00–10:00 a.m.), shading the device to avoid sunlight interference. In each leaf, three readings were taken and the values were averaged.

### 2.8. GA_3_ Analysis Pretreatment Methods

For freeze-drying (IlShinBioBase Co., Ltd., Donducheon, Republic of Korea), leaf sample (30 DAT) 0.2 g homogenized with a homogenizer was taken in a conical tube (SPL Life Science Co., Ltd., Pocheon, Republic of Korea) and 10 mL of purified water was added, and then the pH was adjusted to 3 using 96% sulfuric acid: water (1:1, *v*/*v*). After adding 20 mL of ethyl acetate, it was shaken at 300 rpm for 15 min with a shaking mixer, centrifuged at 4000 rpm for 5 min, and supernatant (ethyl acetate layer) was taken, and this process was repeated twice. After mixing the extracted ethyl acetate layer, 10 mL of pH 8.0 phosphate buffer was added. After shaking at 300 rpm for 15 min with a shaking mixer, centrifugation at 4000 rpm for 5 min. After that, an aqueous layer was collected, and the pH was adjusted to 3.0 using 96% sulfuric acid: water (1:1, *v*/*v*). After adding 10 mL of ethyl acetate (Tokyo Chemical Industry Co., Ltd., Tokyo, Japan) and shaking at 300 rpm with a shaking mixer (Hankuk S&I Co., Ltd., Hwaseong, Republic of Korea) for 15 min, the ethyl acetate layer was taken and the organic solvent volatilized with nitrogen at 45 °C to dry it completely. Then, added was 300 µL of tetrahydrofuran (THF, Sigma-Aldrich Co., St. Louis, MO, USA), 5 µL of triethylamine and 40 µL of 0.2 M phenacyl bromide (Tokyo Chemical Industry Co., Ltd., Tokyo, Japan). After mixing with a vortex mixer for 2 min, it was reacted at 90 °C for 1 h. After completion of the reaction, the test tube was cooled to room temperature and then completely dried at 40 °C. After dissolving the residue with 500 µL of methanol (JT Baker, Phillipsburg, NJ, USA), it was transferred to a vial for HPLC analysis (Shimadzu Co., Kyoto, Japan and model: NEXRA XR).

### 2.9. Tuber Yield Performance of Potato Plant

Potato tubers were manually harvested (90 DAT) and sorted to determine the total yield. After sorting, the tubers were counted and weighed to determine the tubers’ number (TN) per plant, mean tuber fresh weight (TFW), and grade the tuber by weight <1 g, >1 g, and >3 g.

### 2.10. Statistical Analysis

Statistical analysis was conducted using Statistics 10 (Tallahassee, FL, USA), and a one-way analysis of variance was performed. All the results were presented as the mean ± SD (standard deviation). The mean differences were compared by Tukey’s post hoc multiple comparison test. *p* values < 0.05 were significant. Principal component analysis (PCA) was carried out using the OriginLab 10.0 software (OriginLab, Northampton, MA, USA).

## 3. Results

### 3.1. Plant Morphological Characteristics of Potato Plant Grown under LED Light

Table 2 and Table 3 demonstrate the plant morphological characteristics of potato plants grown under different light spectrums. In this study, treatment L3 recorded the best shoot length at 87.3 ± 2.05 (Table 2). However, treatment L6 recorded the lowest (61.34 ± 3.39) (Table 2). Treatment L4 recorded the best performance in the maximum parameters compared to the control (L9) and other light treatments, including the stem diameter (11.08 ± 0.25), leaf number (25.32 ± 1.2), leaf width (19 ± 0.81), root length (49 ± 2.1), and stolon length (49.62 ± 2.05), except the plant fresh weight (67.16 ± 4.06 g), plant dry weight (4.46 ± 0.08 g) and shoot length (Table 2 and Table 3). Whereas, a significantly minimal branch number was recorded in treatments L1, L6 and L2 (1.33 ± 0.47, 1.66 ± 0.41 and, 2.33 ± 0.47, respectively) (Table 3) Moreover, treatments L2, L8, and L9 also increased the shoot length significantly. In addition, treatments L4 and L8 significantly increased the underground root and stolon morphology as well. The results also showed that treatments L1 and L2 significantly increased the stem length, but in the rest of parameters, their performance was minimal. However, under the L5 and L6 treatments, the plants expressed significantly lower morphological parameters and biomass accumulation (Table 2 and Table 3).

### 3.2. Potato Tuber Yield

L4 resulted in a significant enhancement in the tuber number, demonstrating a 51% improvement compared to the control (L9). However, the highest tuber biomass (73 g) was observed in the control treatment (L9) (Figure 2). The results further revealed that treatment L4 predominantly influenced the production of larger tubers (>3 g, >1 g), with values of 9.26 ± 3.01 and 11.69 ± 2.14, respectively. Consequently, this treatment exhibited a noteworthy 13.6% increase in the production rate of larger tubers (>3 g) compared to the control. On the other hand, L5 and L6 demonstrated minimal performance in producing larger tubers (2.00 ± 0, 3.01 ± 1.01) and (2.21 ± 1.29, 3.84 ± 1.06), respectively, (>3 g, >1 g), as recorded in Figure 3.

### 3.3. Photosynthetic Gas Exchange Measurement

The photosynthetic response changed in the different light treatments (Figure 4). The photosynthetic rate, transpiration rate, stomatal conductance, and water use efficiency in the current study range were 2.38–3.95 µmol m^−2^ s^−1^, 0.56–6.03 mol m^−2^ s^−1^, 0.003–0.47 mol m^−2^ s^−1^ and 0.65–2.92 µmol m^−2^ s^−1^, respectively. Treatments L1 (3.51 ± 0.06), L4 (2.59 ± 0.05) and L9 (3.95 ± 1.4) have manifested significantly higher (*p* ≤ 0.05) photosynthetic rates compared to the others. Moreover, L1 (3.42 ± 0.18 mol m^−2^ s^−1^, 0.21 ± 0.02 mol m^−2^ s^−1^) and L9 (6.03 ± 0.12 mol m^−2^ s^−1^, 0.47 ± 0.02 mol m^−2^ s^−1^) had the highest state of transpiration and stomatal conductance, whereas minimum results were recorded from L6 (0.56 ± 0.32 mol m^−2^s^−1^, 0.03 ± 0.00 mol m^−2^ s^−1^) and L7 (0.8 ± 0.09 mol m^−2^ s^−1^, 0.02 ± 0.00 mol m^−2^ s^−1^). On the other hand, the maximum WUE was recorded in L6 (3.18 µmol m^−2^ s^−1^) and L7 (3.77 µmol m^−2^ s^−1^).

### 3.4. Effects of Different LED Light Treatments on Chlorophyll Content

The change in the Chl *a*, Chl *b*, TCL, Car, and SPAD values of the potato leaves. Higher Chl *a* was recorded in L1 (0.096 ± 0.00 mg g^−1^), L2 (0.096 ± 0.00 mg g^−1^), L4 (0.093 ± 0.00 mg g^−1^), L7 (0.097 ± 0.00 mg g^−1^) and L8 (0.093 ± 0.00 mg g^−1^), whereas L4 influenced the higher Chl *b* (0.06 ± 0.00 mg g^−1^) and TCL content (0.15 ± 0.00 mg g^−1^). On the other hand, higher Car was observed in treatments L1 (2.95 ± 0.02 mg g^−1^) and L4 (2.88 ± 0.06 mg g^−1^), while a higher SPAD value was recorded from both the L4 (49.28 ± 1.14) and L3 (46.16 ± 1.18) treatments (Figure 5).

### 3.5. Effects of Different LED Light Treatments on Gibberellic Acid (GA3) Content

The GA_3_ content of potato plants grown under different light spectral treatments. From the results, treatment L4 (38.61 ± 0.39 mg g^−1^) was recorded with a significantly higher GA_3_ content, whereas the minimum value was recorded in L5 (17.72 ± 0.49 mg g^−1^) (Figure 6).

### 3.6. Principal Component Analysis (PCA)

For the potatoes of plants grown under LED light conditions in an aeroponic system, all the data were employed to perform PCA (Figure 7). These findings represent initial breakthroughs in the cultivation of potato seed tubers using artificial light. They provide a basis for the development of an advanced LED lighting system specifically designed for growing potato seed tubers under artificial light conditions. Lines starting from the ballot’s center show negative or positive correlations between distinct light treatments. The degree of correlation with tuber production in an aeroponic system in the PCA is determined by their proximity to a specific procedure. Overall, the potato plant under L4 = red: blue: green (70 + 10 + 20) demonstrated optimal outcomes for potato growing, establishment, and production of seed tubers in aeroponic environments.

## 4. Discussion

### 4.1. Plant Morphological Characteristics

The diverse combinations of the artificial LED light spectra have a significant impact on potato plant growth and seed tuber development. Numerous studies have been conducted to determine the optimal dose of red, blue, green, far-red, and white light spectra for plant growth and development. Previous findings have indicated that the optimal red and blue LED light ratio must be specified and varied based on the plant species. For example, plants exposed to a combination LED light spectrum of red and blue light had more photoreceptor activation and photosynthetic activity than plants exposed to monochromatic red or blue light [21]. In this context, blue and red lights are thought to be absorbed primarily on the surface of the leaves by the palisade tissue, while green and far-red lights are penetrated deeply beneath the leaf surface into the foliage [22,23,24]. We observed that the addition of green light combined with red, blue, and white spectra had a positive influence on plant morphological growth. Similar trends were also observed in an earlier study, where green LEDs along with red and blue LEDs increased the most plant morphological parameters, including the leaf area of potato plantlets. This finding also suggested that in addition of green LEDs, the combined red and blue spectra reduced the amount of blue and likely alleviated the stem elongation inhibition significantly [11].

### 4.2. Potato Tuber Yield and Grading

Varied light spectra exert differential impacts on photosynthesis, morphogenesis, and, ultimately, the growth and development of plants [25,26]. The tuberization process. Previous studies narrated that there was no indication that light effects tuberization [27]; rather, hormonal signals, particularly gibberellins (GA) and cytokinin (CK), regulate it [28]. Furthermore, the light-mediated endogenous plant hormones’ regulation, such as GA, ABA, and IAA, has already been documented [29]. In our study, the present findings comply with the previous findings. The GA has been identified as an important plant hormone that regulates photoperiodic-mediated tuberization in the potato and has been shown to induce stolon formation via longitudinal cell expansion via transverse orientation of microtubules and microfibrils to the cell axis [27]. A recent study found higher bioactive GA content in plants grown under red LED than those grown under other light conditions [29]. However, another study recorded that red light inhibited tuber initiation [27], which partially supports the current findings. In the current study, a higher tuber number was recorded in treatment L4 = red: blue: green (70 + 10 + 20), whereas the lowest number was recorded in L5 = red: blue: far-red (70 + 25 + 5). These findings indicate a negative influence of red + far-red light on the tuberization process. Therefore, the lower tuber number and tuber weight in the present study could be the result of red and far-red light influencing phytochromes, as the involvement of phytochromes in the regulation of potato tuberization was previously hypothesized [30]. In our current study resembling these findings, L4 = red: blue: green (70 + 10 + 20) and L5 = red: blue: far-red (70 + 25 + 5), from the basis of the previous study we can narrate that green light has a positive influence on tuberization when added with (red + blue) compared to far-red light (Figure 3). Moreover, the results show that the combination of red, blue, green as well as red + blue, far-red and red and far-red light had a significant impact on stem elongation, which eventually led to tuber formation. Another hormone, indole acetic acid (IAA), is thought to improve the ability of plant organ sinks [31]. It has been demonstrated that red light increases the IAA concentration in potatoes, thereby promoting the flow of assimilates into tubers [32]. The rate of assimilation of assimilates is also an important factor in tuber size and weight [33]. Increased assimilation rates are efficiently partitioned into underground tubers for plants growing in the combined LED blue and red range. This could explain why the majority of large microtubers have been found in the red–blue spectrum [32,34,35]. These findings partially support our current study, as L4 (red + blue + green) had a positive effect on tuber size.

### 4.3. Photosynthetic Gas Exchange Measurement

The stomatal density, distribution, and opening status have a significant impact on photosynthesis because they regulate water vapor diffusion and carbon dioxide uptake in plants. Furthermore, many factors, such as light and temperature, can influence stomatal behavior [36]. It is also known that the photosynthetic rate is dependent on the chlorophyll content and can be affected by its diversity [37,38]. These findings support our study, as L4 manifested comparatively a higher chlorophyll content, photosynthetic rate, and stomatal conductance (Figure 4 and Figure 5). The efficacy of blue and red LEDs is higher than that of white and green LEDs [39]. Furthermore, green light plays a vital role in photosynthesis because it aids plant adaptation to varying light intensity [14]. These findings completely support our study, as treatments L1 and L4 having red, and blue with different concentrations of green light increased A and E. Green light can boost up photosynthesis activity combined with high white light more effectively than red light. The plant surface reflected green light, yet penetrating green light stimulated photosynthesis with high efficiency [40]. It was also narrated that green light penetrates down plants and deeper into leaves, resulting in more uniform light absorption throughout the leaf and supplying excitation energy to cells further away from the adaxial surface. This can boost leaf photosynthesis when the PPFD is high [14,22,23].

### 4.4. Photosynthetic Pigments

Chlorophyll is an essential element for light absorption. Consequently, the photosynthetic capacity is increased through chlorophyll content increases [41,42,43]. Blue light is readily absorbed by the photosynthetic antenna of plant pigment, which acts as a catalytic agent in the accumulation of photosynthetic pigment in plant leaves [44,45]. A severe malformation of the chloroplast of in vitro potato plantlets was recorded when grown under a monochromatic red light only; however, a combination of red and blue light provided more uniform chloroplasts in the leaf with increased thickness of the leaves [46]. When the red and blue spectra were combined, the photosynthetic pigment was greatly increased compared to other spectrum combinations [32]. In addition, a higher percentage of green with red and blue LEDs has evidence of more chlorophyll accumulation [47]. According to a previous study, the LED light treatment with red + blue + far-red had a positive influence on the chlorophyll content [9]. This result supports our study, as treatments L7 and L8 containing more far-red light increased the chlorophyll content (Figure 5).

### 4.5. Effects of Different LED Light Treatments on Gibberellic Acid (GA_3_) Content

Gibberellin (GA) hormones are a grouping of phytohormones that regulate many aspects of plant growth and development [48,49]. One of the most important aspects of GA function is its role in regulating the growth rate of plant tissues or organs by influencing cell proliferation and expansion [49,50]. Plants have various mechanisms that harmonize hormone-driven processes, such as modulating hormone synthesis, transporting, and signaling, because hormones essentially contribute to plant growth and development [51]. These processes, which include numerous hormonal pathways, are frequently light-regulated to stimulate plant growth and development. Earlier studies revealed that under the red LED light, the rice seedlings had the highest GA_3_ content to promote cell division. As a result, the rice seedlings grown under the red LED light treatment had the higher plant height and fresh weight [52,53]. These findings partially support our current study, as L4 (red + blue + green) had a significant effect on the GA_3_ content in potato plant growth. Some studies demonstrated that the plants were less responsive to GA when the monochromatic red light is used, but opposite results were recorded in case of blue light. This is likely because red light inhibits positive GA signaling even when higher GA levels are present [54]. However, despite having a high ratio of red light, our study found higher content of GA_3_ in several treatments, like L2, L3, L4, and L8 (Figure 6). This variation might be the result of other LED light combined with red light, which has a direct effect on GA_3_ production, as we discussed earlier.

## 5. Conclusions

According to the current findings, a higher tuber number, tuber size (>3 g), and GA_3_ content, along with better plant growth characteristics, were exhibited by treatment L4, a light spectrum combined with red: blue: green (70 + 10 + 20) LEDs. In addition, the photosynthetic pigments and photosynthetic activity were found to be better in the L1, L4, and L7 treatments. Considering plant photo-morphological and tuberization performance, treatment L4 was found with the best spectral composition compared to all the treatments.

## Figures and Tables

**Figure 1 plants-13-00737-f001:**
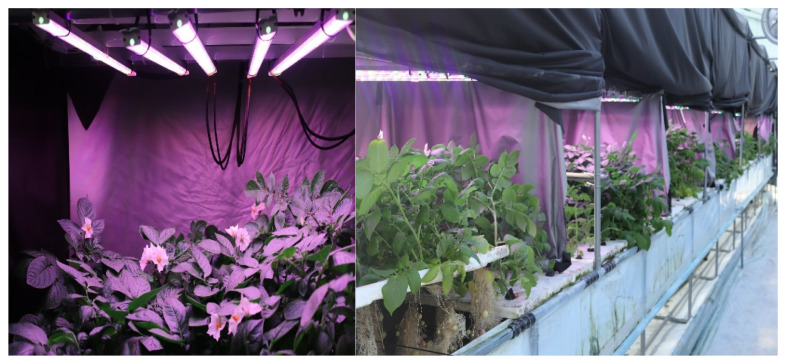
Photographs of potato plants grown under different artificial LED light spectrums.

**Figure 2 plants-13-00737-f002:**
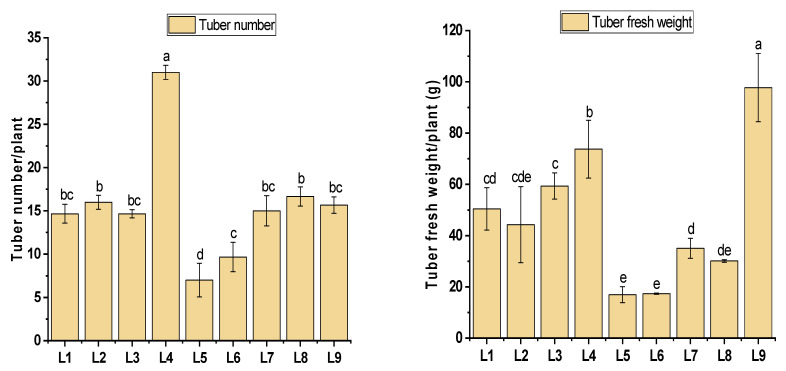
Tuber yield of potatoes grown under different LEDs light spectra in the aeroponic culture system. Significant differences (*p* < 0.05) are indicated by different letters in each bar graph. L1 = red: blue: green (70 + 25 + 5), L2 = red: blue: green (70 + 20 + 10), L3 = red: blue: green (70 + 15 + 15), L4 = red: blue: green (70 + 10 + 20), L5 = red: blue: far-red (70 + 25 + 5), L6 = red: blue: far-red (70 + 20 + 10), L7 = red: blue: far-red (70 + 15 + 15), L8 = red: blue: far-red (70 + 10 + 20), L9 = natural light.

**Figure 3 plants-13-00737-f003:**
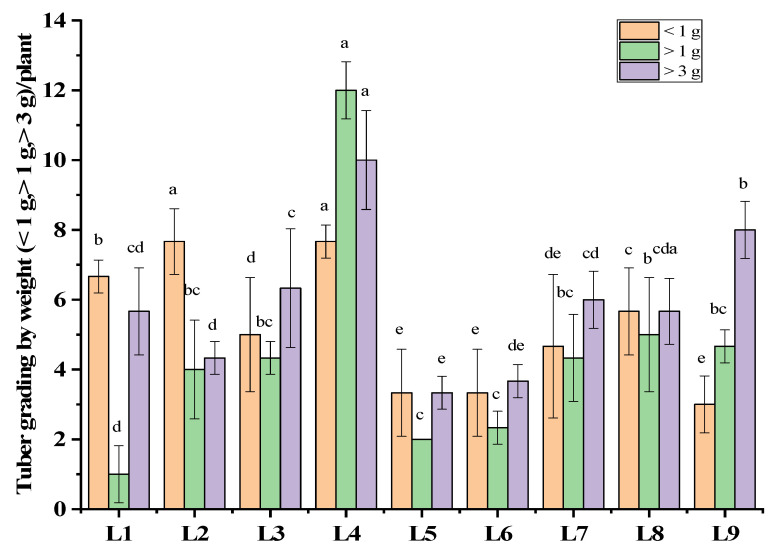
Tuber grading of potato grown under different LEDs light spectra in the aeroponic culture system. Significant differences (*p* < 0.05) are indicated by different letters in each bar graph. L1 = red: blue: green (70 + 25 + 5), L2 = red: blue: green (70 + 20 + 10), L3 = red: blue: green (70 + 15 + 15), L4 = red: blue: green (70 + 10 + 20), L5 = red: blue: far-red (70 + 25 + 5), L6 = red: blue: far-red (70 + 20 + 10), L7 = red: blue: far-red (70 + 15 + 15), L8 = red: blue: far-red (70 + 10 + 20), L9 = natural light.

**Figure 4 plants-13-00737-f004:**
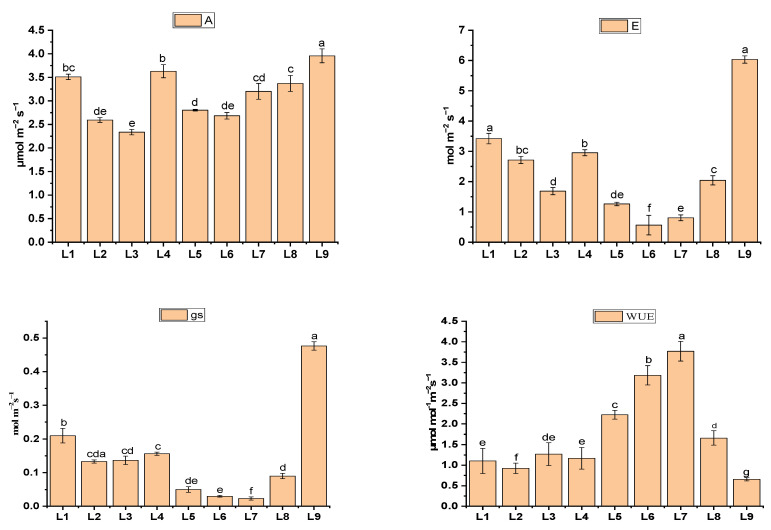
Net photosynthetic rate (A), transpiration rate (E), stomatal conductance (gs), and water use efficiency (WUE) of potato plants grown different LEDs light spectra in an aeroponic culture system. Significant differences (*p* < 0.05) are indicated by different letters in each bar graph. L1 = red: blue: green (70 + 25 + 5), L2 = red: blue: green (70 + 20 + 10), L3 = red: blue: green (70 + 15 + 15), L4 = red: blue: green (70 + 10 + 20), L5 = red: blue: far-red (70 + 25 + 5), L6 = red: blue: far-red (70 + 20 + 10), L7 = red: blue: far-red (70 + 15 + 15), L8 = red: blue: far-red (70 + 10 + 20), L9 = natural light.

**Figure 5 plants-13-00737-f005:**
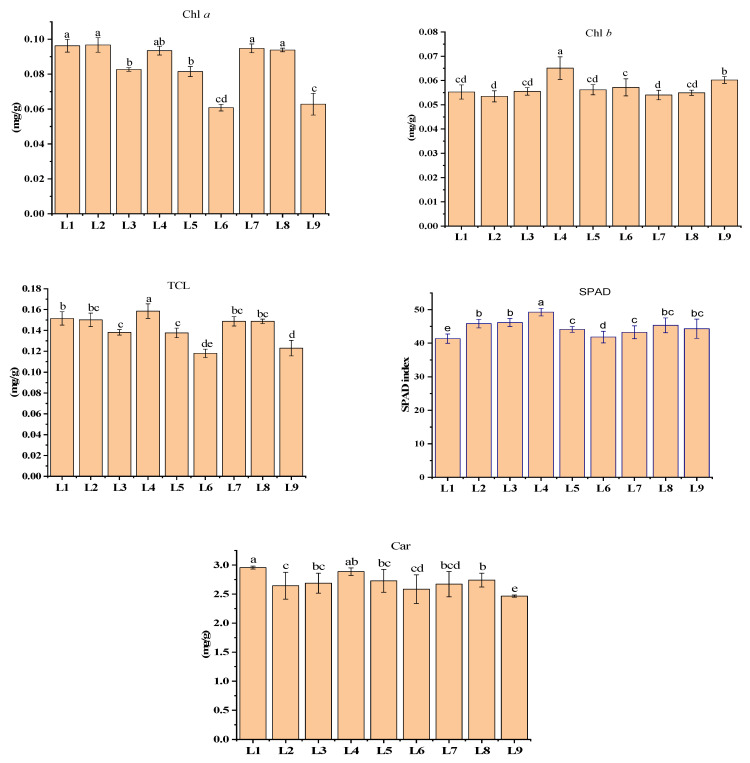
Chlorophyll *a* (Chl *a*), chlorophyll *b* (Chl *b*), total chlorophyll (TCL), carotenoid (car) and SPAD index of potato plants grown under different LED light spectra in an aeroponic culture system. Significant differences (*p* < 0.05) are indicated by different letters in each bar graph. L1 = red: blue: green (70 + 25 + 5), L2 = red: blue: green (70 + 20 + 10), L3 = red: blue: green (70 + 15 + 15), L4 = red: blue: green (70 + 10 + 20), L5 = red: blue: far-red (70 + 25 + 5), L6 = red: blue: far-red (70 + 20 + 10), L7 = red: blue: far-red (70 + 15 + 15), L8 = red: blue: far-red (70 + 10 + 20), L9 = natural light.

**Figure 6 plants-13-00737-f006:**
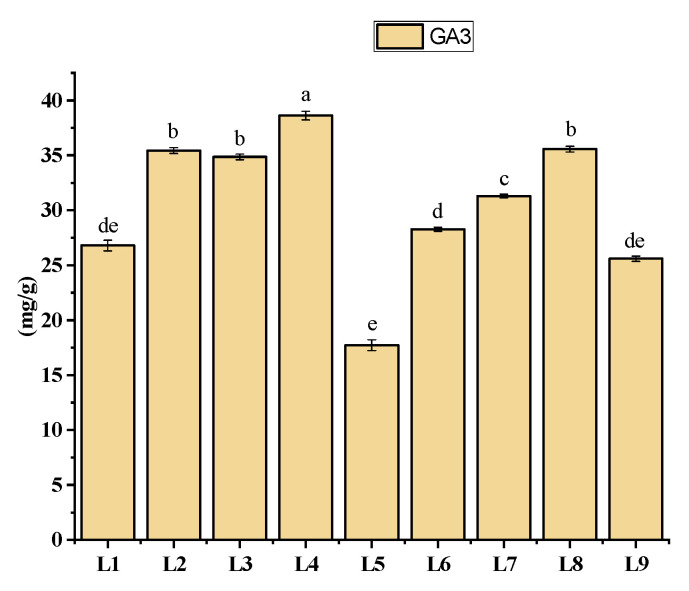
The GA3 content of potato plants grown in an aeroponic system. Significant differences (*p* < 0.05) are indicated by different letters in each bar graph. L1 = red: blue: green (70 + 25 + 5), L2 = red: blue: green (70 + 20 + 10), L3 = red: blue: green (70 + 15 + 15), L4 = red: blue: green (70 + 10 + 20), L5 = red: blue: far-red (70 + 25 + 5), L6 = red: blue: far-red (70 + 20 + 10), L7 = red: blue: far-red (70 + 15 + 15), L8 = red: blue: far-red (70 + 10 + 20), L9 = natural light.

**Figure 7 plants-13-00737-f007:**
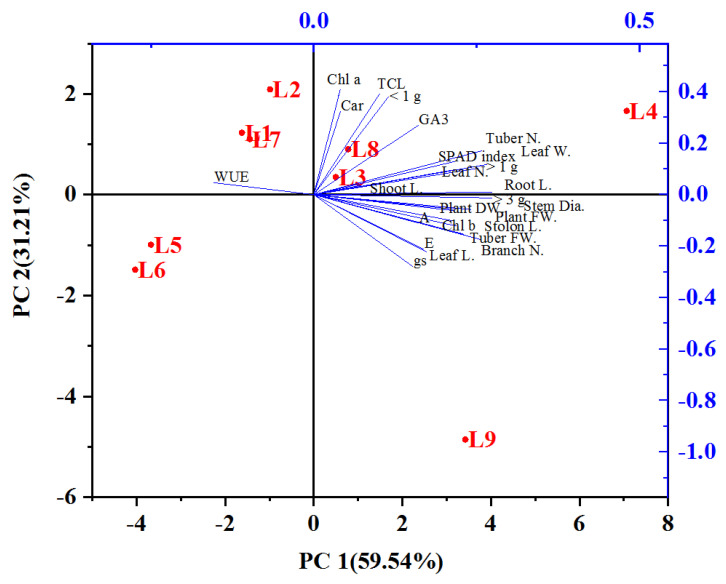
Patterns and associations between treatments are represented by principal component analysis (PCA). Stem L. (Stem length); stem dia. (stem diameter); leaf N. (leaf number); leaf L. (leaf length); leaf W. (leaf width); branch N. (branch number); root L. (root length); stolon L. (stolon length); PFW (plant fresh weight); PDW (plant dry weight); A (photosynthetic rate); E (transpiration rate); gs (stomatal conductance); WUE (water use efficiency); Chl *a* (chlorophyll *a*); Chl *b* (chlorophyll *b*); Tch (total chlorophyll); Car (carotenoid); SPAD index; GA3 (gibberellic acid content) tuber N. (tuber number); tuber FW. (tuber fresh weight); tuber N. (tuber number); TFW (tuber fresh weight); <1 g, (less than 1 g); >1 g, (more than 1 g); >3 g (more than 3 g).

**Table 1 plants-13-00737-t001:** LED light spectrum combinations using eight LED light treatments on potato plants; fraction of integral photon flux.

Lights *	Spectrum Combinations *	Intensity (µmol m^−2^ s^−1^) **	Code Name
Red: blue: green	70 + 25 + 5	300	L1
Red: blue: green	70 + 20 + 10	300	L2
Red: blue: green	70 + 15 + 15	300	L3
Red: blue: green	70 + 10 + 20	300	L4
Red: blue: far-red	70 + 25 + 5	300	L5
Red: blue: far-red	70 + 20 + 10	300	L6
Red: blue: far-red	70 + 15 + 15	300	L7
Red: blue: far-red	70 + 10 + 20	300	L8
Natural light		257 **	L9

* red (660 nm); blue (450 nm); green (520 nm); far-red (730 nm). ** The light ratio and intensity were determined using a PG200N handheld spectral PAR meter (UPRtek, 165 Vogt 21, Aachen 52072, Germany).

**Table 2 plants-13-00737-t002:** The shoot length, stem diameter, leaf number, leaf length and leaf width of plants grown under different LED light spectra in the aeroponic culture system.

	Shoot Length (cm)	Stem Diameter (mm)	Leaf Number	Leaf Length (cm)	Leaf Width (cm)
L1	82.66 ± 2.86 bc	7.37 ± 0.31 d	18 ± 0.8 bc	22 ± 0.81 d	13 ± 0.81 c
L2	84.3 ± 3.29 b	7.61 ± 0.37 d	20.66 ± 0.4 b	21.6 ± 1.2 e	14 ± 0.81 bc
L3	87.67 ± 2.05 a	8.23 ± 0.2 c	20 ± 1.63 b	24.66 ± 0.94 c	15.61 ± 1.24 b
L4	70 ± 1.63 d	11.8 ± 0.25 a	25.32 ± 1.2 a	27.65 ± 1.24 b	19 ± 0.81 a
L5	69.34 ± 2.49 d	8.56 ± 0.48 c	15 ± 1.1 cd	27.6 ± 1.24 b	10 ± 0.81 e
L6	61.34 ± 3.39 e	7.83 ± 0.12 d	12.67 ± 0.81 e	21 ± 0.81 e	11.33 ± 0.47 d
L7	74 ± 3.26 c	7.03 ± 0.13 de	16.3 ± 0.94 c	22.3 ± 1.24 d	14 ± 0.81 bc
L8	84 ± 2.94 b	8.79 ± 0.21 c	13 ± 0.81 e	25.36 ± 0.47 c	14.6 ± 1.24 bc
L9	84.3 ± 3.85 b	9.50 ± 0.55 b	19.66 ± 1.63	29 ± 0.81 a	15 ± 0.81 b

Significant differences (*p* < 0.05) are indicated by different letters in each value represented in the table. L1 = red: blue: green (70 + 25 + 5), L2 = red: blue: green (70 + 20 + 10), L3 = red: blue: green (70 + 15 + 15), L4 = red: blue: green (70 + 10 + 20), L5 = red: blue: far-red (70 + 25 + 5), L6 = red: blue: far-red (70 + 20 + 10), L7 = red: blue: far-red (70 + 15 + 15), L8 = red: blue: far-red (70 + 10 + 20), L9 = natural light.

**Table 3 plants-13-00737-t003:** The branch number, root length, stolon length, plant fresh weight and plant dry weight of plants grown under different LED light spectra in the aeroponic culture system.

	Branch Number	Root Length (cm)	Stolon Length (cm)	Plant Fresh Weight (g)	Plant Dry Weight (g)
L1	1.33 ± 0.47 e	28 ± 2.16 de	23.3 ± 1.22 e	41.35 ± 0.85 de	2.91 ± 0.02 cd
L2	2.33 ± 0.47 d	31.3 ± 1.22 cd	25.3 ± 3.68 e	43.48 ± 0.68 d	2.93 ± 0.04 cd
L3	4.32 ± 1.24 b	37.68 ± 1.6 bc	41.63 ± 2.86 bc	55.47 ± 2.58 bc	3.26 ± 0.1 c
L4	13.3 ± 0.94 a	49 ± 2.1 a	49.62 ± 2.05 a	67.16 ± 4.06 ab	4.46 ± 0.08 b
L5	2.3 ± 0.47 d	23.6 ± 1.69 e	30.33 ± 1.25 de	38.38 ± 0.99 e	2.72 ± 0.05 cd
L6	1.66 ± 0.41 e	29 ± 0.88 d	34.33 ± 1.68 d	44.4 ± 1.19 c	2.87 ± 0.02 cd
L7	3.31 ± 0.33 c	32 ± 1.66 c	37.6 ± 1.3 c	59.84 ± 3.26 b	3.59 ± 0.07 c
L8	3.33 ± 0.36 c	36 ± 0.88 bc	40.3 ± 0.44 bc	72.69 ± 2.28 a	5.02 ± 0.09 a
L9	13 ± 0.82 a	39.6 ± 1.22 b	43 ± 0.89 b	67.4 ± 1.53 ab	4.48 ± 0.04 b

Significant differences (*p* < 0.05) are indicated by different letters in each value represented in the table. L1 = red: blue: green (70 + 25 + 5), L2 = red: blue: green (70 + 20 + 10), L3 = red: blue: green (70 + 15 + 15), L4 = red: blue: green (70 + 10 + 20), L5 = red: blue: far-red (70 + 25 + 5), L6 = red: blue: far-red (70 + 20 + 10), L7 = red: blue: far-red (70 + 15 + 15), L8 = red: blue: far-red (70 + 10 + 20), L9 = natural light.

## Data Availability

Data are contained within the article.

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
