# Peer review of "Effect of Light Quality on Seed Potato (Solanum tuberose L.) Tuberization When Aeroponically Grown in a Controlled Greenhouse"

_plants, 2024, doi:10.3390/plants13050737_

Round 1

Reviewer 1 Report

Comments and Suggestions for Authors

Line 66     Vigorous…. capital letter

Line 115       m2-)and….with space character

Line 289   (11,26) and following lines  ….with space character

Tab. 2 … too much information, hard to read, need new formation

Fig. 2    better as bar graph like Fig 3

Fig. 4    axis lettering should be not in the middle of the graph, like Fig. 5

Fig. 5   to small, hard to read axis lettering

Fig 7   way to small and not to read, should be mentioned at the paper text before

Comments on the Quality of English Language

need some minor work, see my comments

Author Response

Comments and answers

Line 66     Vigorous…. capital letter

Answer: Fixed. Line 66.

Line 115       m2-)and….with space character

Answer: Fixed.  Line 115

Line 289   (11,26) and following lines  ….with space character

Answer: Fixed. Lines (11,26,289),

Tab. 2 … too much information, hard to read, need new formation

Answer: The table has broken down in two. Lines 218 and 219

Fig. 2    better as bar graph like Fig 3

Answer: Fixed. Fig. 2 has changed. Lines 264

Fig. 4    axis lettering should be not in the middle of the graph, like Fig. 5

Answer: Fixed. Fig. 4 have changed. Line 303.

Fig. 5   to small, hard to read axis lettering

Answer: Fixed. Fig. 5 has changed. Line 363.

Fig 7   way to small and not to read, should be mentioned at the paper text before

Answer: Fixed. Fig. 7 has changed. Line 486

Reviewer 2 Report

Comments and Suggestions for Authors

The manuscript by Rahman et al studied the effects of different light quality on potato growth and tuber development, and revealed that treatment L4 is the best combination of light quality, which has important guiding significance and application value for seed potato production. There are several issues that need to be revised:

1. Since L9 was used as the control (line 29), L1-L8 should be compared with L9 in the results section, instead of saying “positive influence” in general (line 192), at least the definition of “positive influence” should be explained. If L9 is used as a control, the fresh and dry weights of L4 do not differ significantly from L9 (line 194). In the same way, L4 produces fewer tuber biomass than L9, so the word “increased” is inappropriate (line 209).

2. Were morphological parameters determined 70 days after transplanting (line 204)? The determination process needs to be briefly described in the Materials and Methods section. For example, potato leaves are pinnate compound leaves. In Table 2, leaf number seems to be the number of compound leaves, while leaf length and leaf width seem to be the parameters of leaflets. For root/stolon length, is the total length of all roots/stolon? Or the longest root/stolon length of a plant? Is length measured in millimeters or centimeters? At what point from the base/top of the plant was stem diameter measured? How is a branch defined?

3. Is the sample used for determination of GA3 content the same as that used for determination of chlorophyll (line 156)? The concentration, type and state (free or bound) of gibberellins vary according to the developmental stage of the plant.

4. Although Fig 2 and Fig 4 look nice, the connection diagram (Fig 2) and the axis intersections (Fig 4) do not seem to meet the requirements of an academic paper.

Comments on the Quality of English Language

Moderate editing of English language required

Author Response

Comments and answers

The manuscript by Rahman et al studied the effects of different light quality on potato growth and tuber development, and revealed that treatment L4 is the best combination of light quality, which has important guiding significance and application value for seed potato production. There are several issues that need to be revised:

  1. Since L9 was used as the control (line 29), L1-L8 should be compared with L9 in the results section, instead of saying “positive influence” in general (line 192), at least the definition of “positive influence” should be explained. If L9 is used as a control, the fresh and dry weights of L4 do not differ significantly from L9 (line 194). In the same way, L4 produces fewer tuber biomass than L9, so the word “increased” is inappropriate (line 209).

Answer: Fixed. Please check in lines 29 to 35, Lines 179-193, Lines 202-214,

  1. Were morphological parameters determined 70 days after transplanting (line 204)? The determination process needs to be briefly described in the Materials and Methods section. For example, potato leaves are pinnate compound leaves. In Table 2, leaf number seems to be the number of compound leaves, while leaf length and leaf width seem to be the parameters of leaflets. For root/stolon length, is the total length of all roots/stolon? Or the longest root/stolon length of a plant? Is length measured in millimeters or centimeters? At what point from the base/top of the plant was stem diameter measured? How is a branch defined?

Answer: Fixed. Line180-194

  1. Is the sample used for determination of GA3 content the same as that used for determination of chlorophyll (line 156)? The concentration, type and state (free or bound) of gibberellins vary according to the developmental stage of the plant.

Answer:  GA3 content was determination 30 DAT line 157.

  1. Although Fig 2 and Fig 4 look nice, the connection diagram (Fig 2) and the axis intersections (Fig 4) do not seem to meet the requirements of an academic paper.

Answer: Fixed. Fig. 2 and Fig. 4 have changed. Lines 264 and 303.